# Parameter Identification of Permanent Magnet Synchronous Motor Based on LSOSMO Algorithm

**DOI:** 10.3390/s25092648

**Published:** 2025-04-22

**Authors:** Songcan Zhang, Zhuangzhuang Zhou, Yi Pu, Yan Li, Yingxi Xu

**Affiliations:** 1College of Information Engineering, Henan University of Science and Technology, Luoyang 471000, China; 220320271848@stu.haust.edu.cn (Z.Z.); 230320040621@stu.haust.edu.cn (Y.L.); yingxixu@haust.edu.cn (Y.X.); 2CAMA (Luoyang) Measurements &Controls Co., Ltd., Luoyang 471000, China; puy013@avic.com

**Keywords:** PMSMs, parameter identification, spider monkey optimization algorithm, chaotic mapping, adaptive t-distribution method, opposition-based learning strategy

## Abstract

The exact identification of the parameters of Permanent Magnet Synchronous Motors (PMSMs) is extremely significant to reach servo system’s excellent performance control. So as to solve the problems of slow PMSM parameter identification using the spider monkey algorithm, and easily falling into local optimal and having unstable identification results; the LSOSMO algorithm is put forward in this article, which combines logistic–sine chaotic mapping strategy, dynamic probability adaptive t-distribution method, and an opposition-based learning strategy to determine PMSMs’ electric parameters (stator resistance *R_s_*, dq-axis inductance *L_d_*, *L_q_*, and flux linkage $\psi_{f}$). First, the logistic sinusoidal chaotic mapping strategy was used to enhance the uniformity of the initial population of the spider monkey optimization (SMO) algorithm. Then, in the local leader stage and the local leader decision stage of the SMO, the dynamic probability adaptive T-distribution method and opposition-based learning strategy are used to replace the greedy selection strategy, increase the position disturbance, and balance the global search and local search ability of the algorithm, so as to improve the performance and convergence speed of the algorithm. The simulation results prove that, compared to the other five algorithms’ identification results, the four parameters that are identified by the LSOSMO algorithm exhibit higher stability and accuracy, with errors that are relative to the true values remaining below 1.1%. The effectiveness and reliability of the identification algorithm is further verified by this.

## 1. Introduction

PMSMs are widely applied in electric vehicles [1,2], industrial automation, and robots, owing to its high efficiency, high-pitched power density, and good dynamic performance [3,4]. PMSMs are multivariable and strongly coupled nonlinear devices [5], its operating characteristics are easily affected by ambient temperature, magnetic field saturation and load disturbance [6,7,8]. These results may change the PMSMs’ electric parameters like stator resistance, inductance, etc.; therefore influencing the motor’s functional characteristics and even cutting the stability and control accuracy of the entire servo control system [9].

An accurate real-time acquisition of a PMSM’s internal parameters is crucial for achieving high-performance control [10,11,12]. Consequently, researchers worldwide have extensively investigated parameter identification methods for PMSMs. These methods can be classified into two principal kinds: offline identification [13,14] and online identification [15,16].

Offline identification techniques for PMSMs are mostly partitioned into signal-based approaches and model-based approaches. The model-based method [17] establishes mathematical equations by deriving the motor’s physical model and combining measurement data with system identification algorithms to identify parameters. This method has high accuracy and reliability [18], but it requires complex mathematical models and high real-time requirements [19]. The signal-based method uses signal processing and statistical analysis to identify parameters by obtaining the data generated during motor operation. Though this way is elementary to operate, the accuracy of parameter identification is easily affected by environmental factors [20].

In comparison with the offline identification method, the online identification technique uses the extraneous parameters of the motor like voltage, rotor velocity, and current, to obtain the electromagnetic parameters of the motor [21]. The online identification methods mainly include four types: the model reference adaptive method [15,22,23,24], the extended Kalman filter method [25,26,27], the least squares method [28,29], and the swarm intelligence optimization algorithm [30]. The model reference adaptive method is used for online identification by motor model, but its accuracy is easily affected by model error. The extended Kalman filter (EKF) combines motor models and measured data for parameter estimation, but in nonlinear systems, it is sensitive to initial conditions and model inaccuracies, which limits its practical feasibility. The least square method is used to achieve parameter estimation by reducing the square of the residual difference between measured and predicted values, but this method is very sensitive to measurement errors.

To address the limitations of traditional PMSM parameter identification methods, researchers have explored swarm intelligence optimization algorithms, yielding significant results. A modified fuzzy particle swarm optimization (MDFPSO) method was pro-posed in reference [31]. A parameter identification’s precision is improved by introducing a convergence factor, but at the same time, it is convenient to fall into the local optimal situation. The CGCRAO algorithm in reference [32] achieves high identification accuracy but requires significant computational resources. In reference [33], the author proposes an improved cuckoo search algorithm, which adopts a fuzzy reasoning strategy based on cloud membership to improve the identification speed of parameters. Neural network is applied to identify the parameters of a high-speed PMSMs in reference [34]. Although this method provides high identification accuracy, it requires a large amount of training data, making it cumbersome for users. The authors in reference [35] integrate the dimension learning-based hunting (DLH) optimization strategy into the Grey Wolf Optimization (GWO) algorithm to effectively reduce identification errors; however, the convergence speed of motor parameters identified by IGWO algorithm needs to be upgraded.

The spider monkey optimization algorithm (SMO) [36], inspired by the foraging behavior of spider monkeys, has attracted much attention due to its novelty, strong optimization ability, and fast convergence speed. However, the algorithm has some problems such as the lack of population diversity, poor local search ability and low search accuracy. At the same time, the SMO algorithm is rarely used in PMSM parameter identification.

For these reasons, we try to apply the SMO algorithm to PMSM parameter identification and propose the LSOSMO algorithm to solve the problems of the SMO algorithm easily falling into local optimal and obtaining unstable identification results. First, the logistic–sine chaotic mapping is used instead of the random generation strategy to initialize the population so as to increase the diversity of the population and ameliorate the initial solution’s quality. Second, the dynamic probability adaptive t-distribution and opposition-based learning strategy are used to iteratively update the population, which can not only enhance optimization performance but also convergence speed, while mitigating the tendency of SMO to fall into local optima. Finally, the identification model of PMSMs is established in the synchronous rotation coordinate system of the dq-axis, and the LSOSMO algorithm is used to identify the motor parameters. In comparison with the other five algorithms, the excellence and feasibility of the LSOSMO algorithm are further confirmed.

This paper’s structure is organized as follows: Section 2 provides a brief introduction to the PMSMs’ mathematical modeling. Section 3 elaborates on the principles of the improved LSOSMO algorithm, with a focus on its core equation, population initialization strategy, and iterative updating mechanism. Section 4 introduces the construction of the PMSM’ parameter identification model and details the LSOSMO parameter identification process. Section 5 presents the identification results of the LSOSMO algorithm compared to other algorithms in a simulation scenario. Lastly, Section 6 summarizes the full text, and further proposes the next research direction.

## 2. Mathematical Model of PMSMs

When the eddy current and iron losses of the PMSMs are neglected, the stator voltage equation of the PMSMs in the synchronous rotating reference frame (d-q coordinate system) is expressed as follows in (1):(1)$$\left\{\begin{array}{l} u_{d}=R_{s}i_{d}+L_{d}\frac{di_{d}}{dt}-\omega_{e}L_{q}i_{q}\\ u_{q}=R_{s}i_{q}+L_{q}\frac{di_{q}}{dt}+\omega_{e}L_{d}i_{q}+\omega_{e}\psi_{f} \end{array}\right.$$
where the stator resistance is represented by $R_{s}$, while the dq-axis stator current and voltage are expressed as $i_{d}$, $i_{q}$, $u_{d}$ and $u_{q}$. The term $\omega_{e}$ stands for the rotor’s electrical angular velocity. $\psi_{f}$ indicates the permanent magnet rotor flux linkage, and $L_{d}$, $L_{q}$ refer to the dq-axis inductance.

To facilitate implementation and analysis on computers, the voltage equation of PMSMs in steady state operation can be discretized to obtain the discrete form of the voltage equation as shown in Equation (2):(2)$$\left\{\begin{array}{l} u_{d}(k)=R_{s}i_{d}(k)-\omega_{e}(k)L_{q}i_{q}(k)\\ u_{q}(k)=R_{s}i_{q}(k)+\omega_{e}(k)L_{d}i_{q}(k)+\omega_{e}(k)\psi_{f} \end{array}\right.$$
When the PMSMs enter the steady state, the negative sequence current with ${i_{d}}^{*}\neq 0$ is injected. The current waveform is shown in Figure 1. The full rank fourth-order equations are obtained by combining Equation (2) with the second-order voltage equation generated by injecting negative sequence current. The unique solution of the four parameters is obtained. The final equation of PMSM under the dq-axis identification is shown in Equation (3):(3)$$\left\{\begin{array}{l} u_{d0}(k)=-\omega_{e0}(k){\hat{L}}_{q}i_{q0}(k)\\ \begin{array}{l} u_{q0}(k)={\hat{R}}_{s}i_{q0}(k)+\omega_{e0}(k){\hat{\psi }}_{f}\\ u_{d1}(k)={\hat{R}}_{s}i_{d1}(k)-\omega_{e1}(k){\hat{L}}_{q}i_{q1}(k)\\ u_{q1}(k)={\hat{R}}_{s}i_{q1}(k)+\omega_{e1}(k){\hat{L}}_{d}i_{d1}(k)+\omega_{e1}(k){\hat{\psi }}_{f} \end{array} \end{array}\right.$$
where the subscript 0 denotes variables and parameters when ${i_{d}}^{*}=0$, and subscript 1 denotes variables and parameters when ${i_{d}}^{*}=-2$. The four parameters to be identified include $\left\{{\hat{R}}_{s}{\hat{L}}_{d}{\hat{L}}_{q}{\hat{\psi }}_{f}\right\}$.

## 3. The Design of Improved SMO Algorithm

### 3.1. The Principle of SMO Algorithm

The principle of SMO algorithm is to reduce the competitive pressure among individuals by simulating the social behavior of spider monkey groups and adopting the fission fusion process, thus improving the efficiency of searching for food resources [36]. The algorithm consists of the following stages:

#### 3.1.1. Population Initialization

The population containing N spider monkeys was initialized using Equation (4)(4)$$SM_{ij}=SM_{\mathrm{minj}}+U(0,1)\times (SM_{\mathrm{maxj}}-SM_{\mathrm{minj}})$$
where $SM_{i}\;(i=1,2,3,\cdot \cdot \cdot ,N)$ represents the *i*-th monkey’s D-dimensional vector, and each vector corresponds to a potential solution to the optimization problem. $SM_{\mathrm{minj}}$ and $SM_{\mathrm{maxj}}$ represent the lower and upper bounds of the j-th dimension, respectively, and U(0,1) represents a random number in the interval [0,1].

#### 3.1.2. Local Leader Phase (LLP)

During the Local Leader Phase, each spider monkey adjusts its position based on the collective experience of its team members and local leaders, then calculates the new position’s fitness value. The position of the spider monkey is updated accordingly if the new position’s fitness value is better than that of the previous position. The position update equation is given as follows:(5)$$SMnew_{ij}=SM_{ij}+U(0,1)\times (LL_{kj}-SM_{ij})+U(-1,1)\times (SM_{rj}-SM_{ij})$$
where $SM_{ij}$ denotes the *j*-dimensional component of the *i*-th spider monkey, $LL_{kj}$ denotes the *j*-dimensional component of the *k*-th local leader, and ${\mathrm{SM}}_{\mathrm{rj}}\;$denotes the *j*-dimensional component of the *r*-th monkey in the *k*-th group, where *r* is randomly chosen within the group, and r≠i.

#### 3.1.3. Global Leader Phase (GLP)

During the Global Leader Phase, the position is updated based on the experience of the global leader and the local group members. The equation for updating the position is as follows:(6)$$SMnew_{ij}=SM_{ij}+U(0,1)\times (GL_{j}-SM_{ij})+U(-1,1)\times (SM_{rj}-SM_{ij})$$
where $GL_{j}$ represents the *j*-th dimension of the Global Leader Phase, with j∈1,2,3⋅⋅⋅D and j is chosen at random.

At this stage, the spider monkey algorithm updates the individual position by calculating the probability based on the individual’s fitness value to increase the likelihood of selecting a better individual position. The probability calculation is shown in Equation (7):(7)$$probi=0.9\times \frac{fitness_{i}}{\min\_\mathrm{fitness}}+0.1$$
where $fitness_{i}$ represents the fitness value of the *i*-th spider monkey (SM), and min_fitness denotes the minima fitness value among the individuals in the group. The fitness values of the new and old positions are compared, and the position with the lowest fitness is retained.

#### 3.1.4. Global Leader Learning (GLL) Phase

At this stage, the new global leader’s position is identified using a greedy selection technique, whereby the spider monkey with the lowest fitness is chosen. If the global leader remains unchanged, the global limit count is incremented by one.

#### 3.1.5. Local Leader Learning (LLL) Phase

At this stage, a greedy selection strategy is employed to update the position of the local leader in the group, selecting the spider monkey with the lowest fitness as the leader. If the position of the new leader remains unchanged, the local limit counter is incremented by one.

#### 3.1.6. Local Leader Decision (LLD) Phase

If the number of times that the local leader position is not updated reaches the set value, it is called the local leader limit. When this happens, all members of the population choose to randomly initialize the update position according to the magnitude of the perturbation rate pr or update the position according to the Equation (8). Equation (8) is as follows:(8)$$SMnew_{ij}=SM_{ij}+U(0,1)\times (GL_{j}-SM_{ij})+U(0,1)\times (SM_{ij}-LL_{kj})$$

#### 3.1.7. Global Leader Decision (GLD) Phase

If the global leader position is not updated to a set value called the global leader limit, the global leader will continue to group the population until the maximum number of groups (MG) is reached. While the Global Leader Decision Phase is running, the LLL phase is also started to select the local leader of the group. Once the maximum number of groups is reached, all groups will be combined into one group regardless of whether the global leader position changes.

### 3.2. The Design of LSOSMO

The LSOSMO algorithm proposed in this paper first uses logistic–sine chaotic mapping strategy to improve the initial population quality in the SMO algorithm. Meanwhile, in the LLP and LLD phases of the SMO algorithm, the dynamic probability adaptive t-distribution method and opposition-based strategy are adopted to replace the greedy selection strategy, which can balance the global search and local search ability, and improve the performance and convergence speed of the algorithm.

#### 3.2.1. Logistic–Sine Chaotic Mapping Strategy

Swarm intelligence algorithms usually use random methods for population initialization, which leads to uneven distribution and insufficient diversity of the initial population. The nonlinearity and periodicity of chaotic mapping can generate more complex random sequences and increase population diversity. In this study, the logistic–sine chaotic mapping strategy was integrated into the SMO algorithm to build the initial population [37].

The following is the equation for the logistic–sine chaotic mapping:(9)$$x_{i+1}=\mu \times x_{i}\times (1-x_{i})+(4-\mu )\times \sin(\pi \times x_{i})/4$$
where x represents the population particle solution, and μ denotes the chaos multiplier, which is set to 0.5 in this study.

In order to study the effects of different initialization methods on the individual distribution of the population, random initialization and chaotic mapping were utilized to generate the initial population as shown in Figure 2. Figure 2a presents the distribution of the initial population generated by logistic–sine chaotic mapping, and Figure 2b shows the distribution of initial population used by SMO algorithm. In Figure 2, the initial population generated by logistic–sine chaotic mapping is more evenly distributed in the parameter space, thus improving the solution accuracy and optimization ability of the algorithm.

#### 3.2.2. The Strategy of Dynamic Probability Adaptive T-Distribution

In the LLP of the SMO algorithm, the spider monkey algorithm updates the location of the individual according to Equation (5), which only considers the experiences of the individual and local leader, reducing the search ability of the algorithm in later iterations. To address this limitation, an adaptive t-distribution strategy is introduced to enhance the search performance [38]. This strategy increases position perturbation during the LLP and LLD phase, adjusting the degrees of freedom according to the number of iterations to balance the global and local searches, thereby improving the convergence speed and the algorithm efficiency. The specific equation for position updates is as follows:(10)$$X_{i}^{t+1}=X_{i}^{t}+X_{i}^{t}\cdot t(iter)$$
After applying Equations (5) and (8), the population position is updated using Equation (10). The updated population position, denoted by $X_{i}^{t+1}$, represents the new position after adaptive disturbance based on the t-distribution and will replace the previous position $X_{i}^{t}$ generated by Equations (5) and (8). The degree of freedom parameter t(iter) in the equation of adaptive t-distribution changes dynamically with the number of iterations.

The algorithm employs adaptive t-distribution to perturb the population position, introducing randomness while utilizing current position information, aiding in escaping local optima. The t-distribution approaches a Gaussian distribution gradually, thereby accelerating algorithm convergence, as the number of iterations rises. To avoid increasing the computational burden of applying this strategy to every individual in each iteration, a dynamic selection probability strategy is introduced to regulate the use of the adaptive t-distribution, preserving the algorithm’s advantages. The equation for the dynamic selection probability strategy is as follows:(11)$$p=\omega_{1}-\omega_{2}\times (\max iter-iter)/\max iter$$
where maxiter indicates the maximum number of iterations, and iter denotes the current number of iterations. $\omega_{1}$ determines the upper limit of dynamic selection probability and $\omega_{2}$ controls its range of variation. The literature shows that when $\omega_{1}=0.5$ and $\omega_{2}=0.1$, the tuning effect is the best.

The change curve of *p* is illustrated in Figure 3. In the early iteration of the algorithm, the random number is easily larger than the dynamic probability *p*, at which time the algorithm uses the adaptive t-distribution to update iteratively. In the later iteration of the algorithm, as *p* slowly becomes larger, the random number is easily smaller than the dynamic probability *p*. At this time, the algorithm adopts the greedy selection strategy to update iteratively. The dynamic selection probability strategy can reduce the computational burden of each individual in the algorithm and improve the performance of the algorithm.

#### 3.2.3. The Strategy of Opposition-Based Learning

According to the related theories of probability theory, a random solution and its inverse solution are more likely to approach the global optimal solution than two independent random solutions, so the reverse learning strategy is more likely to accelerate the algorithm convergence.

In this study, the opposition-based learning strategy is integrated into the LLP and LLD phase of the SMO algorithm to improve the convergence speed and optimization ability of the algorithm. The opposition-based learning strategy adopted in this paper is shown in Equation (12):(12)$$x^{\prime }=rand(N,D)\times (Lb+Ub)-x$$
where N represents the population size of the spider monkey algorithm, D denotes the dimensionality of the population, while Ub and Lb correspond to the upper and lower bounds of the particles, respectively.

After the SMO algorithm executes the LLP and LLD phase, a new population M is obtained by using Equation (12) and merged with the population generated by Equation (10). Then, the merged population is sorted according to its fitness value. The first N optimal individuals are selected as the final population.

## 4. PMSM Parameter Identification Based on LSOSMO

### 4.1. Parameter Identification Principle of PMSMs

The multi-parameter identification problem of PMSMs can be regarded as a system optimization problem. The parameter identification process of PMSMs is to obtain the adaptive value after the fitness function evaluation of the actual value of the system and the output value of the identification model. The parameters to be identified in the identification model are continuously optimized by the optimization algorithm, and finally the PMSM parameters can be identified [39].

The framework for PMSM parameter identification based on the LSOSMO algorithm is illustrated in Figure 4. Figure 5 presents the structure of the PMSM parameter identification model.

The designed fitness function is shown in Equation (13):(13)$$\begin{array}{c} f(x)=\sum_{t+1}^{n}(\omega_{1}{\left(u_{d0}(t)-{\hat{u}}_{d0}(t)\right)}^{2}+\omega_{2}(u_{q0}(t)-{\hat{u}}_{q0}(t){)}^{2}\\ +\omega_{3}(u_{d1}(t)-{\hat{u}}_{d1}(t){)}^{2}+\omega_{4}(u_{q1}(t)-{\hat{u}}_{q1}(t){)}^{2}) \end{array}$$
where the weight coefficients of the fitness function are denoted as $\omega_{1}$, $\omega_{2}$, $\omega_{3}$, $\omega_{4}$. Even with identical fitness values, varying weight settings can impact the accuracy of parameter identification. Extensive experiments indicate that the highest recognition accuracy is achieved when the weight coefficients are set to 0.25. When ${i_{d}}^{*}=0$, the dq-axis voltage measurements under motor control are denoted as $u_{d0}$ and $u_{q0}$, under the control of ${i_{d}}^{*}=-2$, the estimates obtained from the data collected by the algorithm are represented by ${\hat{u}}_{d1}$ and ${\hat{u}}_{q1}$. Here, $u_{d1}$ and $u_{q1}$ represent the measured values of the motor’s dq-axis voltage obtained when a negative sequence current ${i_{d}}^{*}\neq 0$ is injected, while ${\hat{u}}_{d0}$ and ${\hat{u}}_{q0}$ denote the estimates under the same conditions.

In particular, the deviation between the parameter values of the identification model and the actual values is taken as the input of the fitness function, so that the parameter identification is transformed into an optimization problem. The smaller the fitness value is, the closer the parameter to be identified is to the actual value. With the continuous optimization process of the LSOSMO algorithm, the better PMSM parameter combination to be identified is continuously selected, making the fitness value gradually smaller. When the optimization process is complete, the optimal PMSM parameters can be obtained.

### 4.2. The Flowchart of the LSOSMO in PMSM Parameter Identification

Utilize Equation (9) to generate the initial population, and establish the parameters: Local Leader Limit, Global Leader Limit, and pr, etc. First bullet;Compute the fitness value for each SM by Equation (13).The greedy selection method is employed to determine the positions of LLP and GLP.If the termination condition is not satisfied, execute the following steps: Second bullet;

By Equation (5), a new position is generated, and the optimal solution is identified.The dynamic probability *p* in Equation (11) determines whether to implement the adaptive t-distribution strategy.The opposition-based learning strategy described in Equation (12) is employed to generate the opposition-based learning population.Calculate the fitness value of each individual in the population according to Equation (13) and select the individual with the smallest fitness value as the optimal individual.The selection probability of all group members is calculated according to Equation (7).Use Equation (6) to update the positions of all team members selected by probability ${\text{prob}}_{i}$.Execute steps 1 through 6 to update the local leader position, while the global leader position is updated using the greedy selection strategy.If the local leader position does not update its position within a specified number of times, the group member is redirected to foraging through the local leader decision phase. Generate a new population according to Equation (8) and perform steps 2 to 4 to replace the original population.If the global leader position does not update its position within the specified number of iterations, and the number of subgroups does not reach the maximum groups (MG), the subgroup is divided into smaller groups; otherwise, all groups are merged. First item;Figure 6 illustrates the flowcharts of the LSOSMO.

## 5. Simulation Verification and Analysis

To check the feasibility and effectiveness of the propounded LSOSMO algorithm in the PMSM parameter identification application, the PMSM vector control system model in the dq-axis coordinate system is established in Matlab R2022b/Simulink, as illustrated in Figure 4. The motor parameters of PMSM settings are shown in Table 1.

The LSOSMO algorithm, SMO algorithm, PSO algorithm, GWO algorithm [40], WOA algorithm [41], and HHO [42] algorithm are used to identify the parameters of PMSMs. The parameter settings of each algorithm are illustrated in Table 2. For detailed parameter settings of the GWO algorithm, WOA algorithm, and HHO algorithm, refer to references [40,41,42].

These test algorithms carry out parameter identification under the load torque of 5 N/m and speed of 1000 r/min. In order to avoid accidental factors affecting the stability and accuracy of the experimental results, each algorithm was independently identified 20 times, and the average value of the recognition results of each algorithm was taken as the final output value. The experimental results are listed in Table 3.

In Table 3, the identification average value, identification error, and standard deviation of the algorithm are shown, respectively, and the optimal results of the identification data are shown in bold in the table. In the stator resistance identification, LSOSMO, SMO, PSO, GWO, WOA, and HHO algorithms eventually identified values were 1.0269, 1.0301, 0.9071, 0.7615, 0.8138, and 0.9074, respectively. Compared with the other five algorithms, the identification value of the LSOSMO algorithm is closer to the real value. The identification error and standard deviation of the LSOSMO algorithm are the smallest compared with the other five algorithms, and the identification results are more stable. In the d-axis inductance identification, although LSOSMO and SMO algorithms achieve the same identification effect in terms of identification value and standard deviation and are better than the other four algorithms, the identification error of the LSOSMO algorithm is smaller than that of the SMO algorithm, and its identification accuracy is higher. In the identification of the q-axis inductance, the identification values of the LSOSMO algorithm and the SMO algorithm have the best effect compared with the other four algorithms. Still, in the identification results of standard deviation and identification error, the identification error of the LSOSMO algorithm is the least stable, and the identification result is the most stable. Finally, in the identification of rotor flux linkage, the identification result of LSOSMO is the closest to the real value, the identification accuracy and identification stability are better than other algorithms, and the best identification performance is obtained.

The convergence curves of the fitness values identified by the six algorithms are shown in Figure 7. Compared with the SMO algorithm, the LSOSMO algorithm has higher convergence accuracy and faster convergence speed. At the same time, the LSOSMO algorithm is better than other four algorithms.

The stator resistance identification curve is shown in Figure 8. It can be seen from the figure that the LSOSMO algorithm has a faster identification speed and higher identification accuracy than the SMO algorithm. Moreover, compared with the other four algorithms, the identification results of the LSOSMO algorithm in the later iteration are more stable, and the identification error is much smaller than the other four algorithms.

Figure 9 displays the d-axis inductance identification curve. Compared with the SMO algorithm, the LSOSMO algorithm has a faster convergence speed and smaller identification error. GWO and PSO algorithms easily fall into local optimum, and the identification error is higher than the LSOSMO algorithm. The convergence error of the WOA and HHO algorithm is relatively large. The comprehensive comparison shows that the LSOSMO algorithm has the fastest convergence speed, the smallest identification error, and the best stability in identifying the d-axis inductance parameters.

The q-axis inductance identification curve is shown in Figure 10. Although the six algorithms can converge near the true value in the end, the LSOSMO algorithm can converge quickly and identify the identification value with the minimum error stably.

Figure 11 shows the rotor flux identification curve of a PMSM. As can be seen from the figure, only the LSOSMO algorithm, SMO algorithm, PSO algorithm, and GWO algorithm converge near the truth value, but the LSOSMO algorithm’s identification results, identification speed, and identification stability are significantly better than those of the other algorithms.

In general, the identification accuracy and convergence speed of the four parameters identified by the LSOSMO algorithm in PMSM parameter identification are better than those of the SMO algorithm, and also better than the other four algorithms. It is proven that the improved algorithm is feasible and effective.

## 6. Conclusions

This paper presents an LSOSMO algorithm based on the SMO algorithm, which combines a logical sinusoidal chaotic mapping strategy, dynamic probability adaptive t-distribution strategy, and opposition-based learning strategy. In the initial stage of the algorithm, logistic–sine chaotic mapping is used to enhance population diversity and balance exploration and utilization ability. In the iterative updating stage of the algorithm, the opposition-based learning strategy and dynamic probability adaptive t-distribution strategy are introduced to prevent the algorithm from falling into local optimal, and the convergence speed and efficiency of the algorithm are improved. Simulation results show that the LSOSMO algorithm is faster, more accurate, and more stable than the SMO algorithm, PSO algorithm, GWO algorithm, WOA algorithm, and HHO algorithm. Future work will focus on exploring the performance of the LSOSMO algorithm in parameter identification under different motor working conditions so as to improve the universality of the algorithm in parameter identification of PMSMs.

## Figures and Tables

**Figure 1 sensors-25-02648-f001:**
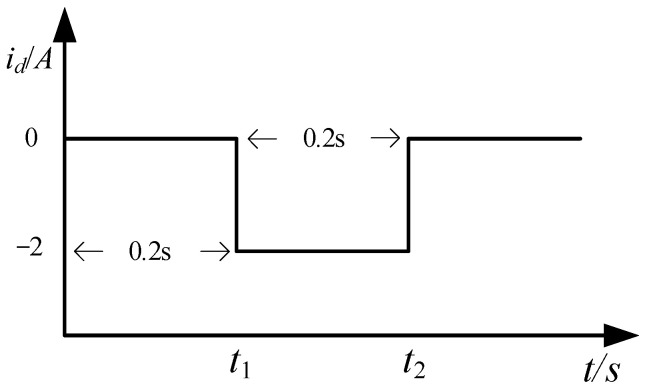
Waveform of the injected d-axis instantaneous negative sequence current.

**Figure 2 sensors-25-02648-f002:**
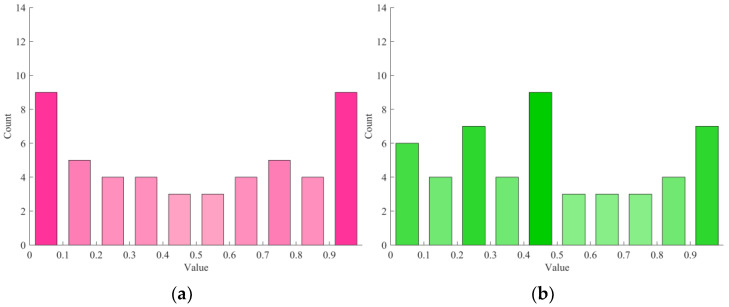
Different diagrams of population distribution. (**a**) Logistic-sine chaos. (**b**) Random.

**Figure 3 sensors-25-02648-f003:**
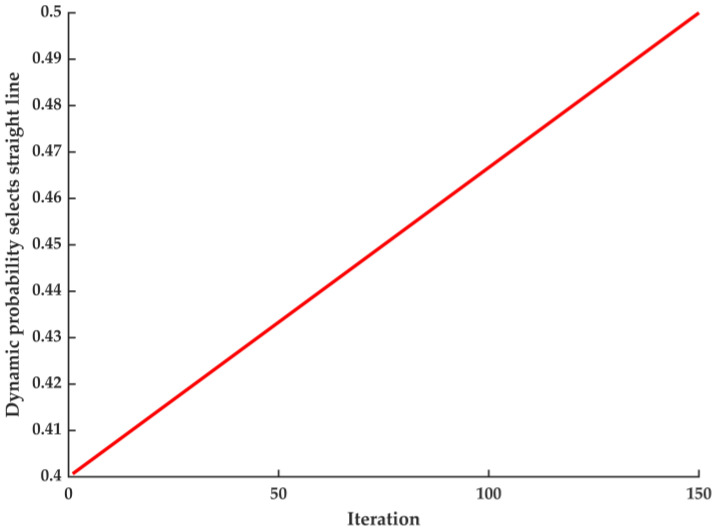
The line of dynamic probability selection.

**Figure 4 sensors-25-02648-f004:**
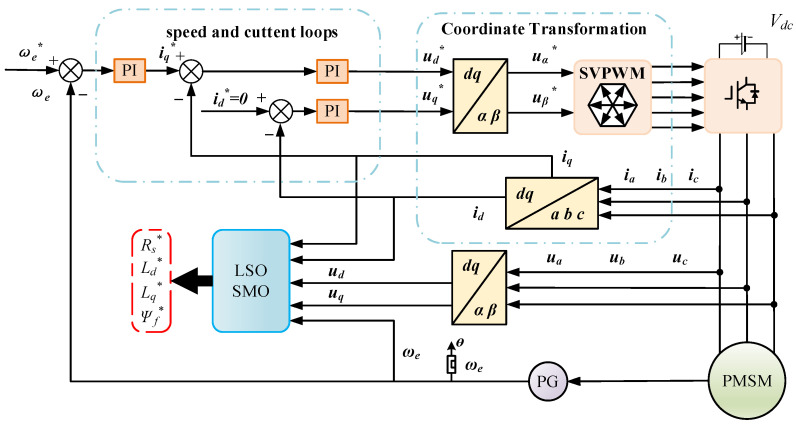
Identification framework of PMSMs based on LSOSMO algorithm.

**Figure 5 sensors-25-02648-f005:**
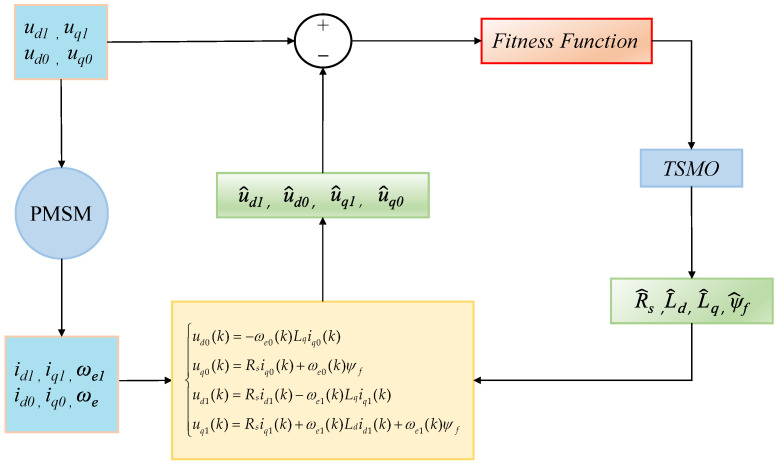
The model of PMSM parameter identification.

**Figure 6 sensors-25-02648-f006:**
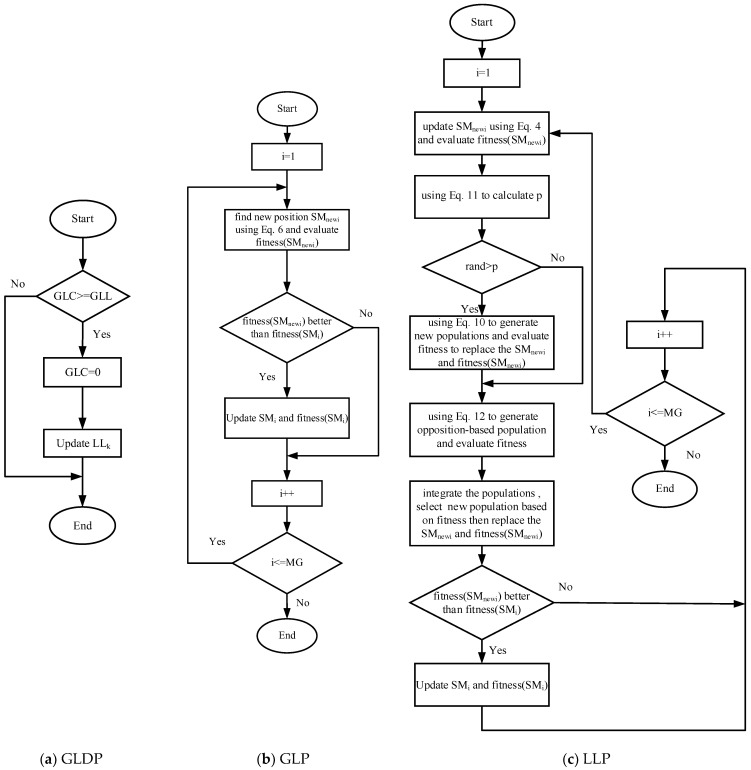
The flowcharts of LSOSMO.

**Figure 7 sensors-25-02648-f007:**
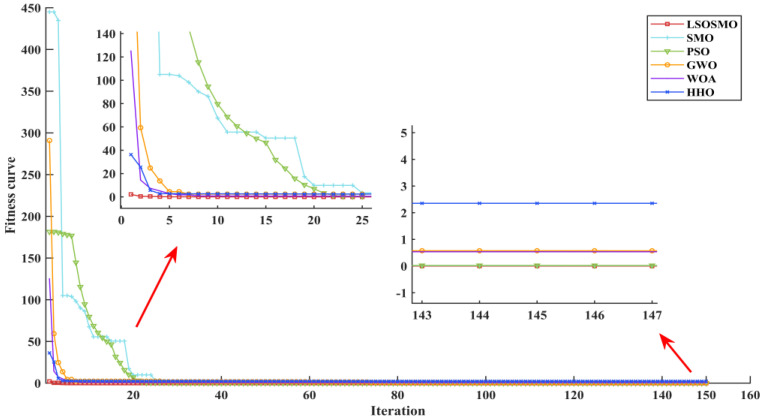
Convergence curve of fitness value.

**Figure 8 sensors-25-02648-f008:**
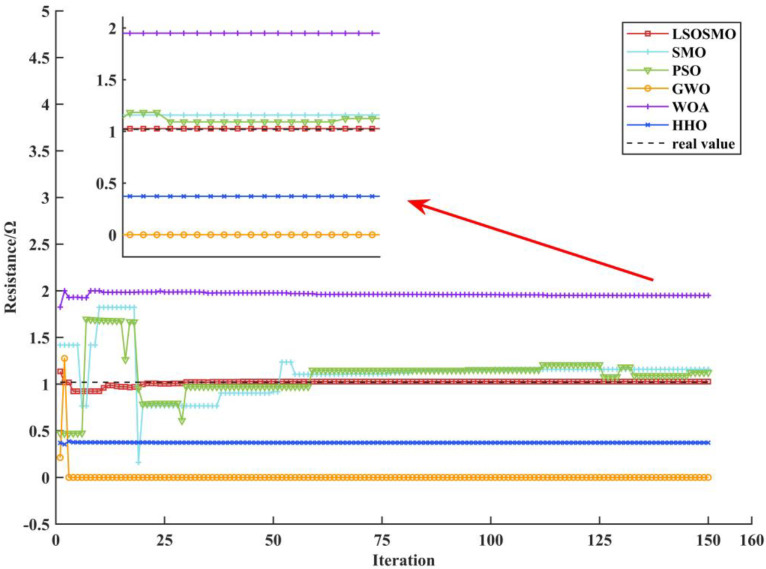
Identification curves for stator resistance.

**Figure 9 sensors-25-02648-f009:**
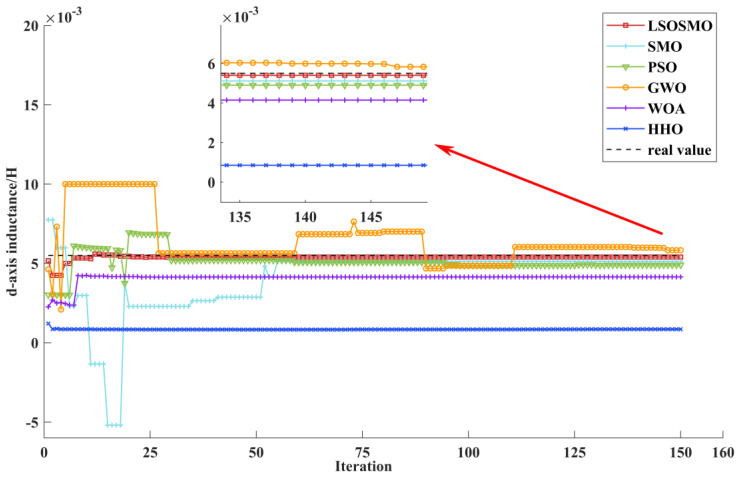
Identification curves for d-axis inductance.

**Figure 10 sensors-25-02648-f010:**
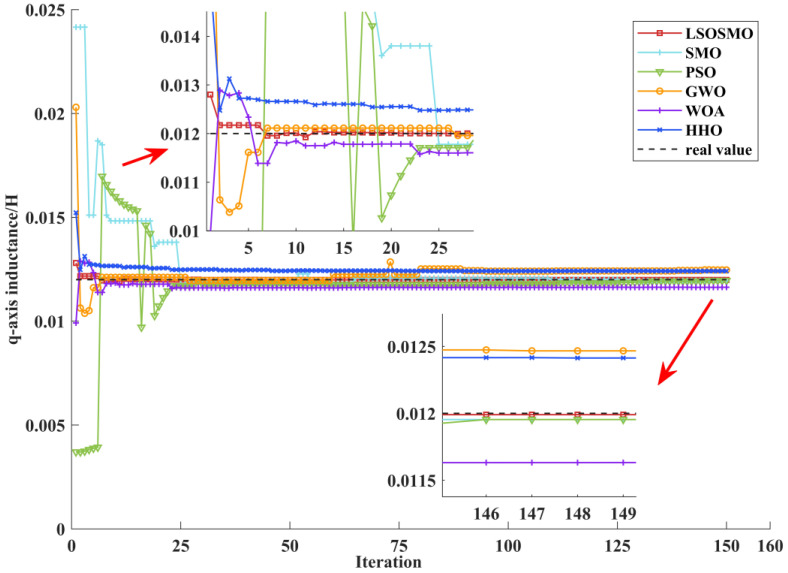
Identification curves for q-axis inductance.

**Figure 11 sensors-25-02648-f011:**
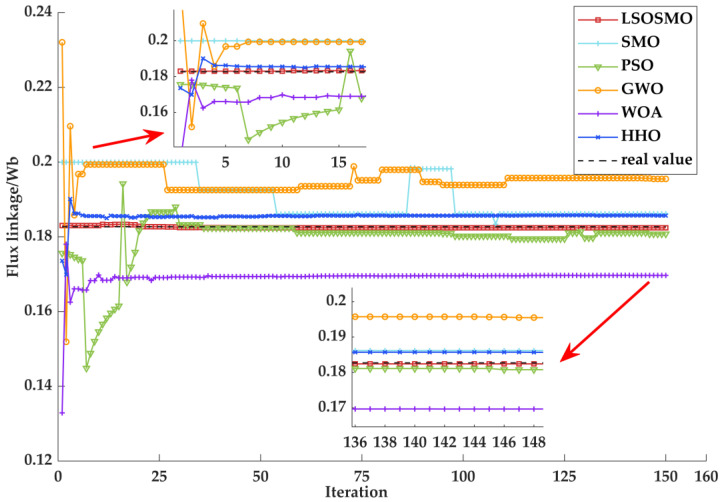
Parameter identification results of flux linkage.

**Table 1 sensors-25-02648-t001:** Specific motor parameters.

Parameter	Value
$R_{s}$/Ω	1.02
$L_{d\;}$/H	0.0055
$L_{q\;}$/H	0.012
$\psi_{f}$/Wb	0.1824
J/(kg·mg^2^)	0.003
B/(N·m·s)	0.008
Polar logarithm P_n_	3
T(N/m)	5

**Table 2 sensors-25-02648-t002:** Parameter Settings of each algorithm.

Algorithm	Parameter Settings
LSOSMO	*pr* = 0.1; μ=4; $\omega_{1}=0.5$; $\omega_{2}=0.1$; *pop* = 50; *N* = 150; MG = 5;
SMO	*pr* = 0.1; *pop* = 50; *N* = 150; MG = 5;
PSO	c1 = 0.8; c2 = 1; $\omega_{\max}=0.9$; $\omega_{\min}=0.4$; pop = 50; *N* = 150;
GWO	*pop* = 50; *N* = 150;
WOA	*pop* = 50; *N* = 150;
HHO	*pop* = 50; *N* = 150;

**Table 3 sensors-25-02648-t003:** PMSM parameter identification results for different algorithms.

Parameter	Value	LSOSMO	SMO	PSO	GWO	WOA	HHO
$R_{s}$	value	**1.0269**	1.0301	0.9071	0.7615	0.8138	0.9074
	error (%)	**0.6754**	0.9904	11.0718	25.3474	20.2115	11.0430
	std	**0.7401 × 10^−4^**	0.8471 × 10^−4^	0.1992	0.4423	0.4661	0.4506
$L_{d}$	value	**0.0054**	**0.0054**	0.0053	0.0053	0.0047	0.0047
	error (%)	**1.0851**	2.1535	3.0015	3.2281	14.6840	14.4618
	std	**0.0014 × 10^−4^**	**0.0014 × 10^−4^**	0.0006	0.0010	0.0018	0.0018
$L_{q}$	value	**0.0120**	**0.0120**	0.0121	0.0121	0.0121	0.0120
	error (%)	**0.0811**	0.1069	0.4248	0.9104	0.6559	0.3945
	std	**0.0004 × 10^−4^**	0.0005 × 10^−4^	0.0001	0.0002	0.0002	0.0002
$\psi_{f}$	value	**0.1824**	0.1823	0.1839	0.1856	0.1843	0.1832
	error (%)	**0.1563**	0.1984	0.6584	1.5863	0.8947	0.2608
	std	**0.0100 × 10^−4^**	0.0105 × 10^−4^	0.0025	0.0059	0.0053	0.0057

## Data Availability

The data presented in this study are available on request from the corresponding author due to privacy.

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
