# Peer review of "Parameter Identification of Permanent Magnet Synchronous Motor Based on LSOSMO Algorithm"

_sensors, 2025, doi:10.3390/s25092648_

Round 1
Reviewer 1 Report
Comments and Suggestions for Authors
Please refer to the attachment.

Reviewer 2 Report
Comments and Suggestions for Authors
Minor revision before publication.

Author Response
Dear Editor:
We are grateful for the opportunity to submit a revised version of our manuscript. We sincerely appreciate the editor and reviewers for their valuable feedback and suggestions on our manuscript entitled “Parameter Identification of Permanent Magnet Synchronous Motor Based on LSOSMO Algorithm” (Manuscript ID: sensors-3510593).
After meticulously incorporating revisions to enhance the clarity, we greatly appreciate the time and effort you dedicated to this review. It is thanks to your guidance and our diligent revisions that we confidently assert the improved quality of content in the revised version. Your constructive criticism and support have been invaluable to us in enhancing our paper. We are committed to aligning the revised version with the highest standards of academic writing.
We have carefully reviewed the reviewers' comments and made revisions to the manuscript. Here are the details:
Comments 1: Please explain the acronym CGCRAO.
Response: Many thanks to the reviewers for pointing out this point. The CGCRAO mentioned in reference 32 cited in this paper uses the abbreviation without the full name description. In reference 32, The author writes, "this paper proposes a CGCRAO algorithm based on chaotic initialization and a hybrid variation strategy. The algorithm uses Tent chaotic mapping for population initialization to improve population diversity. At the same time, by combining the Gaussian and Cauchy distribution characteristics and the three-stage operation idea, the optimal individual variation strategy is autonomously selected in real time to improve the RAO-1 algorithm. " So the CGCRAO algorithm does not have a detailed universal explanation.
Comments 2: Is the LSOSMO algorithm proposed to use offline or online identification?
Response: Thanks for your comments. The LSOSMO algorithm is offline identification.
Comments 3: Please check the grammar after Figure 3. ‘The change curve p….is illustrated instead of illustrat’.
Response: Thanks for your comments. The word "illustrat" in the first sentence after Figure 3 has been modified to "illustrated". On page 7,line 237 .
We apologize for our errors and appreciate the reviewer for assisting us in enhancing the accuracy of our work.
Comments 4: I recommend the appearance of Figure 3 after mentioned in the text.
Response: Many thanks to the reviewers for pointing out this point. I have adjusted the position of Figure 3 after the text in the manuscript.
We apologize for our errors and appreciate the reviewer for assisting us in enhancing the accuracy of our work.
Comments 5: First sentence after Cap. 5. Correct the typing mistake (‘to’ is twice).
Response: Many thanks to the reviewers for pointing out this point. I have removed one "to" in the first sentence after chapter 5. On page 11,line 323.
We apologize for our errors and appreciate the reviewer for assisting us in enhancing the accuracy of our work.
Comments 6: Please correct the typing mistake in the Cap. 6 title.
Response: Many thanks to the reviewers for pointing out this point. I have fixed the typos in the title of chapter 6. It's on page 15, line 397.
We apologize for our errors and appreciate the reviewer for assisting us in enhancing the accuracy of our work.
We would like to express our great appreciation to you and the reviewers for the valuable comments provided on our manuscript, which we have endeavored to revise accordingly. Attached herewith, please find the updated version of the manuscript that we are submitting for your kind consideration. We are eagerly looking forward to receiving your feedback. Thank you for your attention to our work.
Yours sincerely,
Zhuangzhuang Zhou (Second Author)
Corresponding author: Songcan Zhang
E-mail: 220320271848@stu.haust.edu.cn
Date: 2025/4/10

Round 2
Reviewer 1 Report
Comments and Suggestions for Authors
My question has been answered